# Cost-Effectiveness Analysis of COVID-19 Vaccine Booster Dose in the Thai Setting during the Period of Omicron Variant Predominance

**DOI:** 10.3390/tropicalmed8020091

**Published:** 2023-01-30

**Authors:** Kanchanok Sirison, Natthaprang Nittayasoot, Ranida Techasuwanna, Nisachol Cetthakrikul, Rapeepong Suphanchaimat

**Affiliations:** 1Health Intervention and Technology Assessment Program, Ministry of Public Health, Nonthaburi 11000, Thailand; 2Division of Epidemiology, Department of Disease Control, Ministry of Public Health, Nonthaburi 11000, Thailand; 3International Health Policy Program, Ministry of Public Health, Nonthaburi 11000, Thailand

**Keywords:** COVID-19, SARS-CoV-2, COVID-19 vaccines, booster dose, cost-effectiveness, Thailand

## Abstract

The Thai government implemented COVID-19 booster vaccines to prevent morbidity and mortality during the spreading of the Omicron variant. However, little is known about which types of vaccine should be invested in as the booster dose for the Thai population. This study aims to investigate the most cost-effective COVID-19 vaccine for a booster shot as empirical evidence for Thai policymakers. This study applied a stochastic simulation based on a compartmental susceptible-exposed-infectious-recovered model and included system dynamics in the model. We evaluated three scenarios: (1) No booster, (2) A viral vector vaccine as the booster dose, (3) An mRNA vaccine as the booster dose. The incremental cost-effectiveness ratio (ICER) was calculated based on provider perspectives. We found the number of cases in scenarios with viral vector and mRNA booster doses to be lower than in the non-booster group. Likewise, the number of deaths in the viral vector and the mRNA booster scenarios was threefold lower than in the no-booster scenario. Moreover, the estimated grand cost for the no-booster scenario was over 100 billion baht, while viral vector and mRNA scenario costs were 70 and 64.7 billion baht, respectively. ICER shows that viral vector and mRNA scenarios are more cost-effective than the no-booster scenario. Viral vector booster shot appeared to be slightly more cost-effective than mRNA booster shot in terms of death aversion. However, being boosted by an mRNA vaccine seemed slightly more cost-effective than a viral vector vaccine concerning case aversion. In conclusion, policies to promote COVID-19 booster shots in the Thai population by either mRNA or viral vector vaccines are likely to be worthwhile for both economic and public health reasons.

## 1. Introduction

As the COVID-19 pandemic has caused a widespread impact on health and the economy globally, especially among the vulnerable population, it is a critical mission for governments to manage the virus transmission despite epidemiological uncertainties. From the initial wild-type, SARS-CoV2 evolved into various strains, including the Alpha (B.1.1.7), Beta (B.1.351), Gamma (P.1), Delta (B.1.617.2) and Omicron (B.1.1.529) variants, with increased transmission capacities and virulences equal to or greater than the original strain [1,2,3]. Despite the stringent restriction measures, Thailand detected the first case of Omicron in January 2022, soon becoming the majority of COVID-19 infections in the country, accounting for more than 87% of new cases [4].

Thailand has been implementing the COVID-19 vaccine as one of the important measures to alleviate the morbidity and mortality related to COVID-19 infection since February 2021. The first vaccine to arrive was an inactivated vaccine, CoronaVac (Sinovac, Beijing, China), followed by the viral vector ChAdOx1 nCoV-19 vaccine (AstraZeneca, Cambridge, UK), inactivated COVID-19 vaccine (Sinopharm, Beijing, China) and mRNA vaccines (BNT162b2; Pfizer-BioNTech, New York, NY, USA; Moderna, Cambridge, MA, USA) [5]. Through the national immunization plan, the COVID-19 vaccines were quickly distributed and administered, starting with healthcare workers and high-risk groups such as those over 60 years old or those who are immunocompromised, followed by people in the general public [6].

Vaccine efficacy and effectiveness vary by strain, and genetic mutations lead viruses to evade the natural and acquired immunity from vaccination. Many studies suggest the benefit of booster doses against severity and death, especially in the population at high risk of developing serious illness [7,8,9]. Alarmed by the spreading of the Omicron variant in the country, the Thai government rolled out the massive booster dose campaign and urged the population who have received two doses of COVID-19 vaccines to receive a booster (third) shot. Several mix-and-match vaccine regimens were proposed in light of a domestic immunogenicity study showing promising protection. The vaccine effectiveness study suggests an over 80% protection level for the three-dose schedules to prevent COVID-19–associated pneumonia and deaths [10].

According to the current COVID-19 management strategies, the Thai Government has continued to recommend that the population receive up-to-date vaccines, mainly either AstraZeneca or Pfizer-BioNTech, as a booster dose, regardless of any primary vaccine schedules. Since the COVID-19 vaccine will continue to be the main government investment to prevent severe infection and death from COVID-19, the question remains which type of vaccine the government should proceed with as the booster dose policy, and this point covers the objective of this study. The aim of the study is that the evidence generated from it will support the policy decision-making for identifying the most cost-effective COVID-19 vaccine option while maintaining the high preventive benefits for the Thai population.

This article is structured as follows. We started with the introduction (Section 1), as written above. The following section (Section 2) demonstrates the study design and model framework, followed by the model assumption, parameters, formula, outcomes, and ethical considerations. The predicted number of deaths and severe cases of COVID-19 infection and the cost-effectiveness indicators are presented in Section 3. The discussion and conclusion are featured in Section 4 and Section 5, respectively.

## 2. Materials and Methods

### 2.1. Study Design and Model Framework

We employed a stochastic simulation based on a compartmental susceptible-exposed-infectious-recovered (SEIR) model. Most parameters, such as the daily incident cases and incident deaths, were obtained from the internal database of the Department of Disease Control (DDC). Clinical severity data were mainly acquired from the Department of Medical Services (DMS). Basic parameters, such as incubation period and duration of infection, followed recent international literature. Parameters reflecting the healthcare system performance in the country, such as the time lag from being infected to isolation, and current epidemiological force (as identified by reproduction number [R]), were identified by model calibration (as detailed in Section 2.2). The forecasting duration was 90 days, starting from 1 April 2022. Of note is that we used actual daily incident cases and deaths since 15 January 2022 for model calibration (since we assumed that after mid-January 2022, the Omicron variant gained the major share of all variants in Thailand). Thus, the model was run from 15 January 2022 till 30 June 2022 (about 170 days in total).

We also incorporated the idea of system dynamics (SD) in the SEIR framework. The simplified model framework is presented in Figure 1. We divided the entire target population into two groups: (i) The two-dose group (ii) The third-(booster) dose group. The target population in this regard numbered 50 million, equivalent to 70% of the whole Thai population (excluding children under 12 years of age as, at the time of writing, the booster dose policy had not been endorsed for children). In other words, the population scope of this study was Thai people who were potential candidates for the booster dose. In each group, we divided the population into five subgroups: (i) The susceptible, (ii) The exposed, (iii) The infectious before isolation, (iv) The infectious after isolation, (v) The recovered.

The rate of moving from one group (stock) to another, such as from the susceptible group to the exposed group, was determined by a differential equation. The volume of people in each group at any point in time was measured by integrating over the flow. The outflow of the susceptible group was mainly influenced by R. The transition from the exposed group to the infectious group was determined by the incubation period. We split the infectious group into before isolation and after isolation. The time lag before commencing the isolation (either by self-isolation or being admitted to a health facility) was assumed to be about four days (based on model calibration). The recovery time followed the treatment guideline of the DMS. At the time of writing, the target population had all received the second shot. This group then encountered two paths, either remaining as a susceptible person or receiving the third shot. The move from the two-dose group to the three-dose group was governed by the vaccination rate in the population.

### 2.2. Model Assumptions, Parameters, Formula, and Outcomes

We used Microsoft Excel® Version 2211 (Microsoft Corporation, Washington, DC, USA) and Stella version 2.0 (number: 251-401-786-859) (isee systems, Lebanon, NH, USA) for all calculations. The model was run considering the following ideas or assumptions. First, the population of interest was homogenously mixed. All susceptible individuals had an equal chance of being subjected to the COVID-19 infection.

Second, in practice, the exact volume of the infectees at the start of the pandemic could not be known. We proposed that the number of infectees amounted to 0.25% of the population of interest. This assumption corresponded with the experience of the prior waves of COVID-19 cases in Thailand, where the notification of the index case occurred after three to four generations of infection had passed.

Third, there remained a degree of under-reporting, especially for asymptomatic and mildly symptomatic infectees. This phenomenon occurred even in developed countries [11,12]. In Thailand, during the Delta wave in late 2021, the Rural Doctor Society reported that less than one fifth of people in highly populated communities in Bangkok were found infected with COVID-19, but their records were not presented in the official infectee list of the DDC. Therefore, we postulated that the observed asymptomatic or mildly symptomatic cases numbered about one sixth of the actual cases, and the observed non-intubated pneumonic cases constituted about one third of the actual cases. This postulation coincided with the experience of the investigation team and with the opinions of public health experts in the report of the DDC [13]. However, we presumed there was no underreporting of intubated pneumonic cases and deaths.

Fourth, we assumed that the action of the vaccines on the model was represented by two points: (i) The reduction of R (according to vaccine effectiveness [VE] against any infection), (ii)The alteration of the clinical severity profile (according to VE against severe infection and death, causing a greater proportion of asymptomatic or mildly symptomatic cases amongst the third dose vaccinees, relative to those not yet given the third dose).

Fifth, we postulated that the effectiveness of the vaccine did not wane over time, and the immunity level of the vaccinees was boosted up right after the shot without delay.

Sixth, the model calibration was done to estimate many parameters, especially R and the initial infectee volume. We used the “optimisation” function in Stella to calibrate R. The function employed daily reported cases as a payoff to acquire an optimal value of R. Powell’s method was used to find the minimum least square error of the number of daily reported cases.

Seventh, with respect to the stochastic nature of the model, we ran the model for over 50 simulations, with each simulation utilising different values of the incubation period. The incubation period was assumed to follow the Gamma distribution over 1–14 days. Results of the model value at the 2.5th and 97.5th were used to construct a 95% confidence limit.

Eighth, vaccine effectiveness indicators were employed from the domestic nationwide case-control studies using the routine administrative surveillance system of the Vaccine Effectiveness Intelligence Unit of the DDC. At the time of the study, the Omicron strain predominated in Thailand, with a small portion of infectees encountering the Delta strain [10,14].

Ninth, since severe adverse events following immunisation (AEFIs) hardly ever occurs (fewer than 0.85 events per 10,000 doses) and most AEFIs can be recovered without specific treatment, we did not include the cost of AEFIs treatment in the model [15].

Last, we used the third-dose vaccination rate in early 2022 as the fixed vaccination rate throughout the study course. Moreover, the third dose was only administered when an individual was in a susceptible state.

The study centred on the following outcomes: (i) Daily reported incident cases, (ii) Cumulative cases, (iii) Cumulative death toll, (iv) Cumulative grand cost (cost of treatment and vaccination cost). These outcomes were constructed in three scenarios. Scenario 1 was the base (unrealistic, theoretical) situation where the third dose was not administered to the Thai population. Scenarios 2 and 3 were the main focus [14]. In scenario 2, the viral vector vaccine (ChAdOx1 as a proxy) was used as the booster dose. Scenario 3 followed the same idea as scenario 2 but replaced the viral vector vaccine with the mRNA vaccine (BNT162b2 as a proxy). The ultimate outcome was the incremental cost-effectiveness ratio (ICER) based on provider perspectives. We then assessed if scenario 2 or scenario 3 was more cost-effective than scenario 1. ICER was used as a measure to gauge the cost-effectiveness level—the smaller the ICER, the more cost-effective.

The ICER of interest is presented in Table 1. Details of important parameters and formulas of the model are demonstrated in Table 2 and Table 3. The full model structure and details of all equations used are exhibited in Appendix A.

### 2.3. Ethics Consideration

As this study relied only on secondary data from international literature and the internal database of official authorities in Thailand, we were not involved with human participation. Thus, ethics approval was not required. However, the research team followed the principles of ethical standards strictly, as stipulated in the Declaration of Helsinki. No individual information was disclosed.

## 3. Results

We began with an estimation of the incident cases of COVID-19 in Thailand. Figure 2 presents the estimated number of daily new cases in the three hypothetical scenarios (the base scenario with no third-dose booster shot, booster with mRNA vaccine, and booster with viral vector vaccine). All three scenarios showed the same pattern with different magnitudes of COVID-19 cases. The no-booster-dose scenarios showed the highest daily case toll compared with the other two scenarios. If there were no administration of the third dose, the daily cases would peak at 80,892 cases per day (95%CI, 20,065 to 121,972) on day 65, then drop continuously to fewer than 50,000 cases per day by day 90. Scenarios 2 and 3 demonstrate similar results. The viral vector scenario peaks at 28,811 cases per day on day 7, close to the mRNA scenario. The mRNA booster could suppress the daily case toll at 128 cases (95% CI, 5 to 276) on day 170, while the viral vector booster drops to 1510 cases (95%CI, 105 to 3077) at the same time point.

The estimated cumulative number of COVID-19 cases in all scenarios is exhibited in Figure 3. All three scenarios present the same pattern with different accelerations in the rate of growth. The cumulative number of cases of the no-booster scenario is 4,624,833 (95%CI, 3,380,419 to 5,652,603) at the end of day 170. In comparison, the cases in the mRNA booster and viral vector scenarios number 1,867,934 (95%CI, 576,867 to 3,492,275) and 2,331,290 (95%CI, 803,209 to 3,811,999), respectively.

The cumulative number of deaths related to COVID-19 peaks at 4534 in the no-booster scenario (95%CI, 3314 to 5542) on day 170. The mRNA and viral vector show that the volume of deaths is smaller than the no-booster scenario by about threefold. In addition, the mRNA booster scenario keeps the death toll at 1390 (95%CI, 442 to 2584), which is relatively close to the viral vector scenario (1555 deaths, 95%CI, 539 to 2662) (Figure 4).

The cumulative grand cost for scenario 1 (no booster dose) is estimated to reach 109 billion baht (95%Cl, 79,850,246,195 to 133,522,416,727) by day 170. This is far greater than the grand cost in the viral vector scenario (70 billion baht; 95%CI, 34,508,345,033 to 105,054,140,906) and mRNA scenario (64.7 billion baht; 95%Cl, 34,449,983,488 to 102,862,099,871). As the grand cost in both the viral vector scenario and mRNA scenario is far smaller than the no-booster scenario in combination with the lower number of cases and death tolls in the booster scenario relative to the no-booster scenario, the ICER thus shows a negative value—suggesting that the booster scenario, regardless of vaccine types, is cost-saving. The viral vector scenario is slightly more cost-saving (showing more negative ICER) than the mRNA scenario when considering case aversion, whereas the mRNA scenario slightly outperforms the viral vector scenario for death aversion (Table 4).

When focusing on vaccination cost alone (vial cost and administrative cost combined), the mRNA scenario produces an ICER of 6.7 thousand baht per case averted—marginally outperforming the ICER of the viral vector scenario. In contrast, for death aversion, the viral vector scenario demonstrates a more cost-effective result when gauging against the mRNA scenario (5.1 million baht in the viral rector scenario versus 6.5 million baht in the mRNA scenario) (Table 5).

## 4. Discussion

This study provides an insight to policymakers regarding the cost-effectiveness of booster vaccination for the Thai population. The results show that vaccination with mRNA and viral vector vaccines helps reduce the public health burden in terms of the number of COVID-19 cases and deaths. This finding is consistent with other vaccine effectiveness studies. The findings in the study by Kirsebom et al. and Andrews et al. also support the estimated degree of COVID-19 vaccine effectiveness against the Omicron variant. Both studies showed that mRNA vaccines, BNT162b2 (Pfizer–BioNTech) and mRNA-1273 (Moderna), and viral vector vaccine, ChAdOx1 nCoV-19 (AstraZeneca), provided 40% effectiveness in reducing the chance of COVID-19 symptomatic infection and 80% effectiveness against severe infection [20,21]. Our study also expands the existing academic value on this point by suggesting that the booster dose is not only “effective” in preventing infection but also cost-effective and worthwhile for investment over 170 days.

We mainly focus on the cost-effectiveness of mRNA and viral vector vaccines because both vaccine types are recommended by the Thai government to be used as a booster dose. Based on the findings above, with no surprise when considering grand cost, both policy options (viral vector or mRNA booster shot) appeared to favour a cost-saving choice. Therefore, we conclude that both mRNA and viral vector vaccines clearly overshadow the no-booster scenario. In addition, when considering the vaccine cost alone, the mRNA scenario was slightly more cost-effective than the viral vector scenario for case aversion. On the other hand, the viral vector scenario demonstrated a trivially more cost-effective result than the mRNA scenario.

It is worth noting that our model considers just a short period of time. We thus realise that the results are subject to change if we consider re-infection or the drop in vaccine effectiveness when viral mutation occurs [22]. Besides, while the government is on their way to lifting all the restriction measures, this study confirms that the booster dose from either an mRNA vaccine or viral vector can contain the number of COVID-19 cases and deaths. However, despite having high vaccine coverage for early doses in Thailand, vaccine hesitancy on the booster shot is also of critical concern. Many studies affirm some degree of vaccine hesitancy in Thailand, especially among migrants and the elderly [23,24]. This means a massive campaign on the booster shot should continue alongside an adequate supply of vaccines.

Our results are consistent with many studies abroad. For instance, Li et al. pointed out that, compared with two doses of BNT162b2 without a booster, the booster dose (in 100,000 elders in the US) would yield an additional vaccination cost of USD 3.4 million but save about USD 6.7 million in direct medical cost and gain 3.7 quality-adjusted life years (QALY) in 180 days [25]. Fu et al. flagged that administering viral vector booster shots after full immunization (two-dose) of inactivated vaccine amongst the Chinese population would incur an increase of 0.011 QALY with a cost saving of USD 261.7, based on a societal perspective [26].

Despite a thorough analysis, this study is not free of limitations, and the results should be interpreted with caution. First, we relied on many assumptions during the calculation. Moreover, these assumptions were influenced by the health system’s function at that time. For example, the number of COVID-19 cases was retrieved from the COVID-19 surveillance system of the DDC, which mainly captured the cases identified from the reverse transcriptase polymerase chain reaction (RT-PCR) test (as the health facilities were required to report the RT-PCR data to the MOPH). However, the surveillance system cannot fully account for the infected individuals identified by the self-antigen commercial test kit. Moreover, the current DDC guideline does not require a COVID-19 test for all deaths outside health facilities, and even when there is a post-mortem test, a false negative result cannot be ruled out [27]. These are reasons why we attempted to adjust for the underreporting in the model, though we admitted that these problems might not be perfectly solved by model adjustment alone.

Second, as the study used the fixed value of vaccine effectiveness, the results of the model should be carefully interpreted. This is because a vaccine is not the only measure to deal with the force of infection. Social behaviours always play a critical role in disease transmission, and people with different vaccination statuses may have different preventive practices. All these factors are difficult to quantify as quantitative input in the model.

Finally, since the vaccine effectiveness can vary due to the genetic mutation of SARS-CoV-2, the waning of immunity is inevitable if the model considers a long period. As the knowledge of new subvariant is highly dynamic and new subvariants may occur in the future, continuous monitoring and regular assessment of vaccine efficacy and cost-effectiveness are of huge value for both academic and policy reasons.

## 5. Conclusions

COVID-19 booster doses, either with mRNA or viral vector vaccines, are likely to be cost-effective for the Thai population during the Omicron variant transmission. Based on our study, both mRNA and viral vector vaccines remarkably reduce the number of severe cases and deaths from COVID-19 infection. Compared to the no-booster scenario, both mRNA and viral vector booster policies are cost-effective policy options concerning the case and death aversion. In terms of policy implication, the government should maintain an effort to encourage populations, especially vulnerable groups, to receive a booster dose. However, it is worth noting that regular follow-up on vaccine efficacy and cost-effectiveness is recommended.

## Figures and Tables

**Figure 1 tropicalmed-08-00091-f001:**
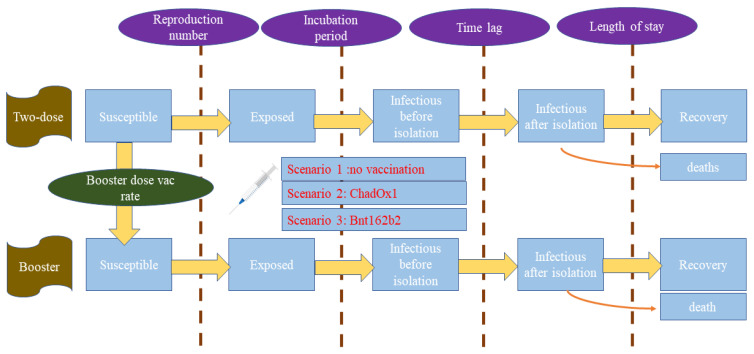
Model framework.

**Figure 2 tropicalmed-08-00091-f002:**
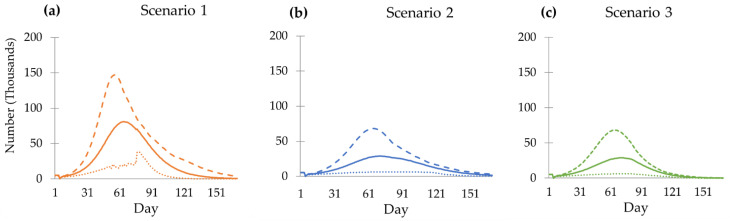
Daily reported incident cases of different scenarios from day 1 to day 170: (**a**) Scenario 1, no booster dose; (**b**) Scenario 2, viral vector vaccine as booster dose; (**c**) Scenario 3, mRNA vaccine as a booster dose. The solid line shows the mean of the outcome. The dashed line shows the 97.5th percentile of the outcome. The dotted line shows the 2.5th percentile of the outcome.

**Figure 3 tropicalmed-08-00091-f003:**
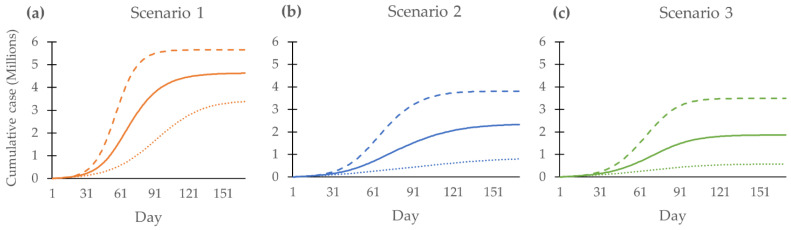
Cumulative cases of all scenarios from day 1 to day 170: (**a**) Scenario 1, no booster dose; (**b**) Scenario 2, viral vector vaccine as booster dose; (**c**) Scenario 3, mRNA vaccine as a booster dose. The solid line shows the mean of the outcome. The dashed line shows the 97.5th percentile of the outcome. The dotted line shows the 2.5th percentile of the outcome.

**Figure 4 tropicalmed-08-00091-f004:**
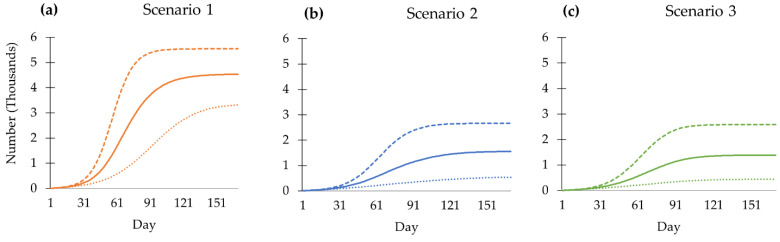
Cumulative deaths of all scenarios from day 1 to day 170: (**a**) Scenario 1, no booster dose; (**b**) Scenario 2, viral vector vaccine as booster dose; (**c**) Scenario 3, mRNA vaccine as the booster dose. The solid line shows the mean of the outcome. The dashed line shows the 97.5th percentile of the outcome. The dotted line shows the 2.5th percentile of the outcome.

**Table 1 tropicalmed-08-00091-t001:** Incremental cost-effectiveness ratio (ICER) of interest.

Comparison	Incremental Cost (a)	Incremental Outcome (b)	Interpretation
Scenario 1 vs. scenario 2	Grand cost 2 − grand cost 1	Cumulative cases 1 − cumulative cases 2	Incremental grand cost per case averted
Scenario 1 vs. scenario 2	Grand cost 2 − grand cost 1	Cumulative death 1 − cumulative death 2	Incremental grand cost per death averted
Scenario 1 vs. scenario 2	Vaccine cost 2 − vaccine cost 1	Cumulative cases 1 − cumulative cases 2	Incremental vaccination cost per case averted
Scenario 1 vs. scenario 2	Vaccine cost 2 − vaccine cost 1	Cumulative death 1 − cumulative death 2	Incremental vaccination cost per death averted
Scenario 1 vs. scenario 3	Grand cost 3 − grand cost 1	Cumulative cases 1 − cumulative cases 3	Incremental grand cost per case averted
Scenario 1 vs. scenario 3	Grand cost 3 − grand cost 1	Cumulative death 1 − cumulative death 3	Incremental grand cost per death averted
Scenario 1 vs. scenario 3	Vaccine cost 3 − vaccine cost 1	Cumulative cases 1 − cumulative cases 3	Incremental vaccination cost per case averted
Scenario 1 vs. scenario 3	Vaccine cost 3 − vaccine cost 1	Cumulative death 1 − cumulative death 3	Incremental vaccination cost per death averted

Note: ICER = (a)/(b) grand cost = cost of treatment + vaccination cost (including vial cost and administrative cost).

**Table 2 tropicalmed-08-00091-t002:** List of key parameters.

Parameters	Unit	Value	Reference (Note)
Reproduction number	Unitless	1.5	Model calibration
Population	Persons	50 × 106	70% of the total Thai population based on the National Statistical Office of Thailand [16]
Mean infectious duration	Days	4.6	Hart et al. (gamma distribution with scale parameter of 0.03 and shape parameter of 165.9) [17]
Mean incubation period	Days	3.5	Helmsdal et al. (gamma distribution with scale parameter of 0.01 and shape parameter of 302.7) [18]
Time lag from being infected to isolation	Days	4	Model calibration
Initial number of infectees	Persons	125,000	Assume 0.25% of the interested population with model calibration
Initial proportion of third dose vaccinees	Unitless	20%	Division of Communicable Diseases, Department of Disease Control
Booster-dose vaccination rate	Persons/day	171,300	Division of Communicable Diseases, Department of Disease Control
Vaccine effectiveness against any infection of viral vector booster dose (two-dose vaccinees as reference)	Unitless	34%	Vaccine Effectiveness Intelligence Unit, Division of Epidemiology, Department of Disease Control
Vaccine effectiveness against any infection of mRNA booster dose (two-dose vaccinees as reference)	Unitless	55%	Vaccine Effectiveness Intelligence Unit, Division of Epidemiology, Department of Disease Control
Vaccine effectiveness against severe infection of viral vector booster dose (two-dose vaccinees as reference)	Unitless	88%	Vaccine Effectiveness Intelligence Unit, Division of Epidemiology, Department of Disease Control
Vaccine effectiveness against severe infection of mRNA booster dose (two-dose vaccinees as reference)	Unitless	78%	Vaccine Effectiveness Intelligence Unit, Division of Epidemiology, Department of Disease Control
Proportion of asymptomatic and mildly symptomatic infectees amongst all infectees	Unitless	99.7%	Internal database of the Department of Disease Control and model calibration (accounted for underreporting factor)
Proportion of non-intubated pneumonic infectees amongst all infectees	Unitless	0.25%	Internal database of the Department of Disease Control and model calibration (accounted for underreporting factor)
Proportion of intubated pneumonic infectees amongst all infectees	Unitless	0.01%	Internal database of the Department of Disease Control and model calibration (accounted for underreporting factor)
Proportion of deaths amongst all infectees	Unitless	0.02%	Internal database of the Department of Disease Control and model calibration (accounted for underreporting factor)
Recovery time for asymptomatic or mildly symptomatic patients	Days	10	Internal database of the Department of Disease Control and model calibration (assume same as clinical profile of the patients during the Delta wave)
Recovery time for pneumonic non-intubated cases	Days	14	Internal database of the Department of Disease Control and model calibration (assume same as clinical profile of the patients during the Delta wave)
Recovery time for asymptomatic or mildly symptomatic patients	Days	21	Internal database of the Department of Disease Control and model calibration (assume same as clinical profile of the patients during the Delta wave)
Recovery time for non-intubated pneumonic patients	Days	21	Internal database of the Department of Disease Control and model calibration (assume same as clinical profile of the patients during the Delta wave)
Administration cost of vaccination	Baht/ dose	234	Meeyai A et al. (3% discount rate adjusted per year) [19]
Viral vector vaccine	Baht/dose	308	Internal database, Division of Communicable Diseases, Department of Disease Control
mRNA vaccine	Baht/dose	488	Internal database, Division of Communicable Diseases, Department of Disease Control

**Table 3 tropicalmed-08-00091-t003:** The essential formulas of the model.

Change of Status	Formula	Note
From susceptible to exposed	−β × (1−VE) × S × I1/P	β = reproduction number/infectious duration, VE = effectiveness of vaccine against any infection, S = susceptible population, I1 = non-isolated infectees, P = total population
From susceptible to non-isolated infectious	−αE	α = 1/incubation period, E = exposed population
From non-isolated infectious to isolated infectious	−δI1	δ = 1/time lag from non-isolation to isolation, I1 = non-isolated infectious population
From isolated infectious to recovered	−ζI2	ζ = 1/length of stay; I2 = isolated infectious population (Varying by severity profile)

**Table 4 tropicalmed-08-00091-t004:** ICER of grand cost per case averted and death averted by day 170.

	ICER of Grand Cost (Thousands) per Case Averted (by Day 170)	ICER of Grand Cost (Millions) per Death Averted (by Day 170)
Scenario 1 vs. scenario 2	−17.0 (95% CI, −17.7 to −16.4)	−13.1 (95% CI, −16.3 to −10.6)
Scenario 1 vs. scenario 3	−16.1 (95% CI, −16.8 to −15.3)	−14.2 (95% CI, −15.9 to −11.2)

Note: Grand cost incorporated both treatment cost and vaccination cost.

**Table 5 tropicalmed-08-00091-t005:** ICER of vaccine cost per case averted and death averted by day 170.

	ICER of Vaccine Cost (Thousands) per Case Averted (by Day 170)	ICER of Vaccine Cost (Millions) per Death Averted (by Day 170)
Scenario 1 vs. scenario 2	7.5 (95% CI, 6.8 to 8.5)	5.1 (95% CI, 4.7 to 5.6)
Scenario 1 vs. scenario 3	6.7 (95% CI, 5.9 to 7.4)	6.5 (95% CI, 6.2 to 7.3)

## Data Availability

Not applicable.

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
