# Peer review of "Cost-Effectiveness Analysis of COVID-19 Vaccine Booster Dose in the Thai Setting during the Period of Omicron Variant Predominance"

_tropicalmed, 2023, doi:10.3390/tropicalmed8020091_

Round 1
Reviewer 1 Report
The article investigates different scenarios of vaccination boosters using a computational model and data from Thailand. The COVID-19 epidemic dynamic is simulated through an SEIR-like model that accounts for two populations, those that received and not received the booster dose. Overall I found the article interesting and I believe that after a major revision it can be accepted for publication. I present below my suggestions and comments.
1 – The English in the text must be carefully revised with particular attention to the Discussion section. In general, I could identify some mistakes, but in the Discussion, the text is hard to read.
2 – The authors must give more details on the SEIR-like model implementation and calibration. The article does not present the system of differential equations that govern the model dynamics. The way the authors address vaccine efficacy is interesting, but they did not mention how they set up the effective rate R in the case of individuals that received the booster dose. It is totally unclear how the model scenarios are simulated. They must show how such “stochastic simulation” works. All this is necessary for reproducibility purposes. Moreover, how sensitive to perturbation are the calibrated parameters? The calibration results must be presented in the text.
3 – The authors should give more details about and justify their assumptions. For example, not considering the observed loss of immunity and reinfection. Since they are interested in shorter periods in their simulations, this can be the main reason for such a choice. Another point is the underreporting issue that they mention. Why did they assume that the total number of actual infections must be six times the reported numbers? How does it affect the disease progression and death toll? There are some articles that provide ways to estimate underreporting using the reported numbers of cases and deaths. See, for example:
(a) Albani, V., Loria, J., Massad, E. et al. COVID-19 underreporting and its impact on vaccination strategies. BMC Infect Dis 21, 1111 (2021). https://doi.org/10.1186/s12879-021-06780-7
(b) Lau H, Khosrawipour T, Kocbach P, Ichii H, Bania J, Khosrawipour V. Evaluating the massive underreporting and undertesting of COVID-19 cases in multiple global epicenters. Pulmonology. 2021;27(2):110–5. https://doi.org/10.1016/j.pulmoe.2020.05.015
4 - In the Results and Discussion sections, when the authors discuss the results considering scenarios 2 and 3, it seems that they mistook their order. Please, revise this part. For example, on page 7, line 190, and page 9, lines 234-235, Scenarios 2 and 3 are in reverse order in comparison to Fig. 2 and Fig 5, respectively.
5 – How was the vaccine efficacy estimated and how does it relate to the effective rate R? How were the other model parameters estimated? The initial number of infectious and exposed individuals must be estimated too. How do the authors find the 95% confidence intervals? How do they generate scenarios? How do the stochastic terms enter the simulations?
6 - Are the results comparable to actual data? More precisely, are the model predictions for the number of deaths for scenarios 2 and 3 similar to the reported data? How was the performance of the implemented vaccination policy in comparison to scenario 1? Is it possible to make such comparisons? If not, the authors must comment on this in the Discussion section.
7 – For reproducibility, the authors must make their codes and the used data available in some public repository.
Author Response
Reviewer 1
The article investigates different scenarios of vaccination boosters using a computational model and data from Thailand. The COVID-19 epidemic dynamic is simulated through an SEIR-like model that accounts for two populations, those that received and not received the booster dose. Overall, I found the article interesting, and I believe that after a major revision it can be accepted for publication. I present below my suggestions and comments.
Comment 1: The English in the text must be carefully revised with particular attention to the Discussion section. In general, I could identify some mistakes, but in the Discussion, the text is hard to read.
Response 1: Thank you for the reviewer comment. We reviewed the English writing and adjusted the language to make it clearer and consistent throughout the manuscript.
Comment 2: The authors must give more details on the SEIR-like model implementation and calibration. The article does not present the system of differential equations that govern the model dynamics. The way the authors address vaccine efficacy is interesting, but they did not mention how they set up the effective rate R in the case of individuals that received the booster dose. It is totally unclear how the model scenarios are simulated. They must show how such “stochastic simulation” works. All this is necessary for reproducibility purposes. Moreover, how sensitive to perturbation are the calibrated parameters? The calibration results must be presented in the text.
Response 2: Full codes of the model and model structure are added in supplementary files 1 and 2. Reproduction number was determined by Powell’s method. Varying the value of incubation period was used to construct stochastic model. We have explained this in more details on the model used in the Materials and Methods Section, please see lines 106-109 and 146-159. Full model structure and equation are demonstrated in Supplementary Files 1-2.
Comment 3: The authors should give more details about and justify their assumptions. For example, not considering the observed loss of immunity and reinfection. Since they are interested in shorter periods in their simulations, this can be the main reason for such a choice. Another point is the underreporting issue that they mention. Why did they assume that the total number of actual infections must be six times the reported numbers? How does it affect the disease progression and death toll? There are some articles that provide ways to estimate underreporting using the reported numbers of cases and deaths. See, for example:
(a) Albani, V., Loria, J., Massad, E. et al. COVID-19 underreporting and its impact on vaccination strategies. BMC Infect Dis 21, 1111 (2021). https://doi.org/10.1186/s12879-021-06780-7
(b) Lau H, Khosrawipour T, Kocbach P, Ichii H, Bania J, Khosrawipour V. Evaluating the massive underreporting and undertesting of COVID-19 cases in multiple global epicenters. Pulmonology. 2021;27(2):110–5. https://doi.org/10.1016/j.pulmoe.2020.05.015
Response 3: We have explained reasons why we used the value of six-fold underreporting. It is mainly due to internal report of the Rural Doctor Society and this value is accepted by the public health experts in the Thai DDC. We are grateful to your advice for the suggested references. We have added this in the reference list. We have added some more discussions on the underreporting phenomenon. Please see lines 130-138.
Comment 4: In the Results and Discussion sections, when the authors discuss the results considering scenarios 2 and 3, it seems that they mistook their order. Please, revise this part. For example, on page 7, line 190, and page 9, lines 234-235, Scenarios 2 and 3 are in reverse order in comparison to Fig. 2 and Fig 5, respectively.
Response 4: Thank you for your comment. We have reviewed the order throughout the manuscript to make sure it is consistent throughout. Scenario 2 was the situation where (since day 78 and onward) all people acquired viral vector vaccine as the booster dose while scenario 3 was the same but with mRNA vaccine.
Comment 5: How was the vaccine efficacy estimated and how does it relate to the effective rate R? How were the other model parameters estimated? The initial number of infectious and exposed individuals must be estimated too. How do the authors find the 95% confidence intervals? How do they generate scenarios? How do the stochastic terms enter the simulations?
Response 5: The vaccine effectiveness values used throughout the paper were retrieved from the Department of Disease Control, Ministry of Public Health, Thailand, as presented in Table 2. We have added more details about model calibration, the stochastic terms, and how 95% confidence interval were constructed in lines 146-159.
Comment 6: Are the results comparable to actual data? More precisely, are the model predictions for the number of deaths for scenarios 2 and 3 similar to the reported data? How was the performance of the implemented vaccination policy in comparison to scenario 1? Is it possible to make such comparisons? If not, the authors must comment on this in the Discussion section.
Response 6: Thank you for the suggestion. However, a direct comparison with the actual data is quite difficult due to certain reasons. First, our model did not include children in the analysis. Second, soon after the paper was finished, the government changed the reporting system by reporting only severe cases. Nonetheless, the rise and fall (pattern) of actual cases is quite similar to our results (peaked by about day 70 then gradually declined).
Comment 7: For reproducibility, the authors must make their codes and the used data available in some public repository.
Response 7: We have added full model structures and equations in supplementary files 1-2 as suggested.

Reviewer 2 Report
Dear Authors,
Thanks for conducting an interesting study and submitting this manuscript to the journal. As you correctly mentioned that there are several limitations in this study and the results should be interpreted with caution, it would be beneficial to Thai government to address hesitancy if you also mention that Covid-19 vaccines reduce severity of the disease symptoms and reduction of hospitalizations.
thanks
Author Response
Reviewer 2
Comment 1:
Dear Authors,
Thanks for conducting an interesting study and submitting this manuscript to the journal. As you correctly mentioned that there are several limitations in this study and the results should be interpreted with caution, it would be beneficial to Thai government to address hesitancy if you also mention that Covid-19 vaccines reduce severity of the disease symptoms and reduction of hospitalizations.
Thanks
Response 1: Thank you for your kind comment. We have discussed about the hesitancy issue in the Discussion section of the manuscript. Please see lines 287-291.

Reviewer 3 Report
Report on “Cost-effectiveness analysis of COVID-19 booster dose using 2 the local surveillance data in Thailand setting” submitted do Tropical Medicine and Infectious Diseases
The article presents a cost-benefit analysis of various booster vaccines and vaccine combinations using the local surveillance data in Thailand. The analysis is based on simulations of SIR models using parameters calibrated using local surveillance data, and thus the results are dependent on the calibrations performed and the specification of the SIR/SEIR model used. In general, I believe that the work has a relevant and very useful contribution in the elaboration of health policies, and I also think that the methodology used is valid as one of the possible tools for this type of analysis, although it has several limitations, as discussed by the authors in the conclusion of the work. So, I am in favor of publishing the article, after incorporating responses to the comments below.
1 – Show the equations of the complete model in an appendix of the work. Although I like the presentation of the model in Figure 1 and Table 3, it is important to have the complete model for sensitivity analysis and reproducibility of results.
2 – The authors should discuss in more detail the procedures for calibrating the parameters contained in Table 2. The results of the work are fully dependent on these parameters. One suggestion would be to perform a sensitivity analysis of the model results in relation to the vaccine effectiveness parameters, for example, using the limits of the confidence intervals of the parameters that were estimated by least squares.
3 – How were the confidence intervals reported in Section 3 calculated?
Author Response
Reviewer 3
The article presents a cost-benefit analysis of various booster vaccines and vaccine combinations using the local surveillance data in Thailand. The analysis is based on simulations of SIR models using parameters calibrated using local surveillance data, and thus the results are dependent on the calibrations performed and the specification of the SIR/SEIR model used. In general, I believe that the work has a relevant and very useful contribution in the elaboration of health policies, and I also think that the methodology used is valid as one of the possible tools for this type of analysis, although it has several limitations, as discussed by the authors in the conclusion of the work. So, I am in favor of publishing the article, after incorporating responses to the comments below.
Comment 1: Show the equations of the complete model in an appendix of the work. Although I like the presentation of the model in Figure 1 and Table 3, it is important to have the complete model for sensitivity analysis and reproducibility of results.
Response 1: We have added full model structures and equations in supplementary files 1-2 as suggested. We also provided more details about the calculation in the main text, please see lines 146-159.
Comment 2: The authors should discuss in more detail the procedures for calibrating the parameters contained in Table 2. The results of the work are fully dependent on these parameters. One suggestion would be to perform a sensitivity analysis of the model results in relation to the vaccine effectiveness parameters, for example, using the limits of the confidence intervals of the parameters that were estimated by least squares.
Response 2: Same as response to Comment #1.
Comment 3: How were the confidence intervals reported in Section 3 calculated?
Responses 3: Same as response to Comment #1.

Author Response
Reviewer 4
Merit of Paper
The manuscript investigates a stochastic simulation based on a compartmental susceptible-exposed-infectious-recovered model and included system dynamics in the model in which the authors evaluate scenarios on the Thai population following three cases of vaccinations. There is any new mathematical theory or tools. But the paper can be used as an experience and interest scientific community in regard to the cost effective of booster vaccination for Thai population.
Comment 1: It will be interesting to present the model considered in the form of a system with stochastic equations. See for example: [1] Mahrouf, Marouane, et al. "Modeling and forecasting of COVID-19 spreading by delayed stochastic differential equations." Axioms 10.1 (2021): 18.
Response 1: Thank you for your kind suggestion. The suggest mentioned paper is quite interesting. We have added more details about the calculation in lines 146-159. However, we did not follow the presentation format of the mentioned paper as we deem our paper is more of policy approach while the mentioned paper focuses more on methodological issues.
Comment 2: Please add in the introduction at their last part, how the paper will be organized.
Response 2: This has been added to the Introduction section, please see lines 73-78.
Comment 3: Correct please: After any author citation, you should add a full stop ‘.’ For example: Replace ‘Kirsebom et al’ by ‘Kirsebom et al.’
Summing up, I recommend the manuscript for publication in the journal after minor revision.
Response 3: We adjusted the author citation as recommended.

Round 2
Reviewer 1 Report
I have no further comments to make.
Reviewer 3 Report
The authors heeded my recommendations, and so my recommendation is for acceptance of the article in its present form.